# The Impact of COVID-19 on Bank Equity and Performance: The Case of Central Eastern South European Countries

**Sylwester Kozak**

Department of Economics and Economic Policy, Institute of Economics and Finance,
Warsaw University of Life Sciences, Nowoursynowska Str. 166, 02-787 Warsaw, Poland;
sylwester_kozak@sggw.edu.pl

**Abstract:** The purpose of this article is to examine the impact of the shock increase, in the value of nonperforming loans, on the equity level and profitability of 141 banks in 18 countries of Central Eastern South Europe (CESE). This study is important for assessing the financial stability of banks in this region in the face of the continuing negative effects of the COVID-19 pandemic. Based on the annual data, as of the end of 2020, from the S&P Global database, stress tests were carried out to check what value of NPL growth, over the next year, will lead to breach the regulatory capital requirements in domestic sectors and in individual groups of banks. The results indicate that the banks in CESE were well capitalized and had the ability to maintain capital requirements with a 12% increase in nonperforming loans. The resilience of domestic banking sectors varies, and it is higher in non-EU countries. Smaller and non-public banks show a greater ability to preserve the appropriate level of equity, although there is a risk that they may postpone the time of provisioning credit risk and additionally increase lending to lower the NPL ratio. Larger banks are more profitable in times of crisis. The results of the research are important for assessing the stability of the banking sector in CESE during the crisis and can be used by financial supervision of the region's countries and banking market analysts.

**Keywords:** COVID-19; banking; credit risk; profitability; CESE countries; emerging economy

## 1. Introduction

The COVID-19 pandemic caused the worst crisis in the global economy since the 2007–2009 global financial crisis. It slowed down, and temporarily froze, the functioning of both the real and financial sectors, including banks. The estimates of the International Monetary Fund (IMF) analysts indicate that the pandemic reduced the value of global GDP in 2020 by 3.2%. GDP fell the most in advanced economies (AE)-by 4.6%, while in emerging market (EM) countries, it fell by 2.1%, including the Emerging and Developing Europe (EDE) countries by 2% as well [1]. The larger losses were experienced by hospitality and tourism industry, retail, and commercial real estate market, as their worldwide sales in the second quarter of 2020 decreased by 80%, 60%, and 50% y/y, respectively [2]. According to some central banks, these negative processes transferred to the financial sector, mainly to banks, and resulted in a significant tightening of the credit policy, as well as a deterioration in the creditworthiness of borrowers, mainly from the SME sector [3–5]. On the customers' side, the uncertainty about the scale of the pandemic development contributed to a decline in demand for financing investment and current capital, as well as consumer goods and services [6].

A 35% drop in the Dow Jones Euro STOXX Banks stock index, between 28 February and the announcement of the pandemic by the WHO on 12 March 2020, as well as a downgrade by Standard & Poor's agency to negative, influenced the credit rating outlook of one-third of banks in EM countries, showing that investors consider banks to be the area particularly exposed to the negative effects of COVID-19 [7]. The banking sector concentrated disturbances experienced by both enterprises and households, including



job cuts, a decline in commercial real estate prices, decrease in liquidity and corporate profitability, drop in central bank interest rates, and an increase in public debt. During the pandemic, the impact of these risk factors was particularly enhanced, and the likelihood of their materialization increased significantly [8]. Among three main types of risk present in the activities of banks, such as credit risk, market risk and operational risk, the credit risk was the highest. For example, in the financial statements of ING Bank N.V., as of the end of 2020, the structure of the capital requirement for risk coverage (in EUR million) was as follows: credit risk–18,948; market risk–714; operating risk–3023; see: ING Bank Additional Pillar III Report 2020; [https://www.ing.com/Investor-relations/Financial-performance/Annual-reports.htm, accessed on 15 August 2021].

The impact of risks on bank performance was usually stronger in EM, as the capitalization of companies and the financial resources of households were smaller than in AE, which created weaker safety buffers for the absorption of unexpected losses [9]. Additionally, a low level of technology and infrastructure made EM more vulnerable to COVID-19. The reduced ability to diversify the revenue structure made the pandemic more detrimental to the financial situation of households and enterprises in EM and limited the ability to service debt. As a result, although the pandemic contributed to a greater short-term decline in GDP in AE than in low-income countries (LIC), according to the World Bank's forecasts in 2021–2022, GDP growth is expected to bounce back to positive numbers in AE but remain negative in LIC [10].

However, the weakening of the COVID-19 impact in the first half of 2021, and the increase in economic activity, did not eliminate the extraordinary risk to which the banking sector is exposed [11]. Many countries important to the world economy, including Australia, Croatia, Greece, Spain, Japan, and the United Kingdom, faced with thousands of daily illnesses, have been forced to re-introduce stringent sanitary restrictions that are likely to reduce the economic viability of enterprises and household incomes. The Financial Stability Board (FSB) reported that, in some countries, the risk rose from the partial closure of public aid programs. Therefore, it suggests the policy of withdrawing public aid be flexible and adjustable to the current situation. The FSB noted that a smooth return to normal operations could also be harmed by the fact that, in the case of some companies, the pandemic disrupted their long-term strategy and forced them to start major restructuring due to the uncertain prospects for the development of the sectors of the economy in which they have operated so far. These actions, even supported by public funds, generate additional credit risk in the long term, as the funds, in whole or in part, have to be repaid in the future [12].

Uncertainty as to the responses of enterprises and, consequently, banks to possible changes in public aid regulations, the necessity to end credit holidays, and further development of the pandemic, motivates some to ask the question: what level of credit losses may lower banks' profits and reduce their equity, so that they will not meet the capital regulatory requirements? Additionally, when assessing the situation in the banking sector, it is important to determine how banks of various sizes, legal status, or locations of business operations are prepared for a shock deterioration in the quality of the loan portfolio.

The study covers 141 banks operating in 18 Central Eastern South European (CESE) countries. The choice of the research topic was motivated by the fact that the studies conducted so far, on the impact of the COVID-19 pandemic on the banking sector, mainly concern North America and the European Union. It is not fully justified to spread the results of these studies to CESE countries. The characteristics of the banking sector in CESE, operating according to the financial intermediation model with the dominance of loans and deposits of the non-financial sector, respectively, in the structure of assets and liabilities, is significantly different than in the AE countries.

The results of the study provide a new insight into the assessment of the resilience of banking sectors in individual countries in the CESE region, as well as individual groups of banks, to shock increases in credit losses, resulting, inter alia, from the further development of the pandemic and legal changes regarding public aid for enterprises and households.

Moreover, they allow us to answer the question of which categories of banks, distinguished by size of assets, quotation on the stock exchange, or geographical area of operation, retain their financial stability longer in times of a financial crisis. The paper is a part of the ongoing research focused on assessing the aftermath of the COVID-19 pandemic in the banking sector. The results of the research are important for analysts of financial supervision and central banks, as well as for academics. They fill the gap in the literature on the subject, shed new light on the problem of assessing the stability of the banking sector in CESE countries during the crisis, and can be used by financial supervision of the region's countries and banking market analysts.

The remaining part of the paper has the following structure. The next section presents a literature review of the impact of the pandemic on the state of economy and the banking sector, followed by data sources and research methodology, as well as modelling results and their discussion. The entire study is summarized in the conclusions.

## 2. Literature Review

A considerable number of studies focused on assessing the impact of COVID-19 on the macroeconomic situation of countries and regions [13], the production volume of enterprises [14], and the level of employment and the situation of micro-enterprises [9,10]. Fernandez, studying the GDP generated in 2020 in Spanish regions, noted that the impact of COVID-19 on the business activities was highly diversified and resulted in GDP decline in the range from 27% in the Balearic Islands to 5.3% in the Extremadura region. Factors worsening the economic situation were, among others: the tourism-focused structure of region's incomes, restrictions on movement of the population, and hiring workers on temporary contracts [13].

In turn, Blanco et al., when examining 8000 Spanish companies, stated that the COVID-19 pandemic has significantly affected their activities, reducing financial liquidity and profitability, which contributed to the greatest decline in economic activity in Spain since 2002 [15]. The decline in companies' revenue and the need to maintain current financial liquidity led to an increase in indebtedness of enterprises, especially in the SME sector. Analysing individual economic sectors, Blanco et al. stated that the largest drops in revenues (measured by gross added value) were recorded by: hospitality and leisure (−24.3%), transportation and storage (−24.1%), and arts, entertainment, and other services (−24%). On the other hand, the smallest losses suffered were, among others, primary sector (+4.8%) and sectors where operations are largely remotely possible, i.e., financial services (+2.9%) and public administration (+1.4%). Additionally, Fernandez et al., based on a survey of 4000 Spanish companies, confirmed that the negative impact of the pandemic was unevenly distributed across regions and sectors [13]. They noted that the largest decreases in revenues and employment were experienced by enterprises with up to 50 employees, as well as enterprises with a short market presence, low efficiency, and a location in urban areas.

Furthermore, Jurgensen et al. indicated there was a diversified impact of COVID-19 on the SME sector. Smaller enterprises frequently have a problem in their operation with the economy of scale, which, in crisis situations, often leads them to insolvency [16]. They are sensitive to changes in demand from both large enterprises, mostly manufacturers [17], and households [18]. Therefore, the prospects of the SME sector for development after the end of public aid vary over time and depend, among others, on the prevailing trends [16]. Changes in banks' lending activity during the COVID-19 pandemic depended on the individual characteristics of banks, as well as the external environment in which they operated. Colak and Oztekin, researching banks in 125 countries, noted that large banks were more exposed to the credit risk, and the scale of such exposition resulted from the structure of the banking market in a country of operation, as well as the regulatory system and institutional environment, the availability of financial markets for enterprises, and the public authorities' response to the health crisis [19]. The strengthening impact of the bank's size, on the increase in credit risk costs caused by the COVID-19 pandemic, was also

confirmed in the study by Borri and di Gorgio, conducted on a sample of listed European banks [20].

Micro and small enterprises were particularly vulnerable to the pandemic crisis and often not fully protected by public aid schemes. The analysis of public aid regulations, related to COVID-19 in OECD member states, showed that the SME sector was not treated on an equal footing with other economic entities and households throughout the period [21]. SMEs, start-ups, self-employed persons, as well as enterprises run by women and enterprises owned by minorities (i.e., ethnic, sexual, and other) were exposed to the risk of receiving insufficient public aid. The final qualification of individual entrepreneurs to obtain governmental aid was subject to certain conditions, while employees employed under a job contract received support almost automatically. The sensitivity of micro and small enterprises to COVID-19 was also confirmed by the research of Cajner et al., which showed that, in the US, in the first quarters of the pandemic, most jobs were lost in small and micro-enterprises employing low-income workers, performing simple jobs, and enterprises providing direct services to the public. This was due to the fact that people with high incomes reduced their expenses on services that could endanger their health, especially in the hotel industry, gastronomy, and tourism. The resulting decline in consumer demand lowered the revenues of service companies and weakened their creditworthiness [22].

Wu and Olson showed that the COVID-19 crisis strengthened the interdependence between banks and the SME sector in China. The deterioration of the financial situation of enterprises increases the costs of risk in banks, while worsening their capital position. Banks have the ability to counteract such a situation, e.g., by granting special lines of credit, lowering interest rates on loans and postponing their repayment, and creating a system of long-term financing of enterprises [23].

Another area generating increased credit risk, stemming from the COVID-19 pandemic, was the labour market. The study by Chetty et al. 2020 found that, for all US workers between March and October 2020, people in the first earnings quartile needed several months to find a job again, while only a few weeks were needed in the case of those in the fourth quartile [8]. The pandemic motivated both entrepreneurs, especially from the service sector, and employees to reconfigure organization of work and business meetings according to the "remote mode". This process led to the liquidation of a significant number of outlets directly serving customers [24].

Aid programs, although they financially supported enterprises and their employees, did not fully compensate for the income lost by COVID-19. The losses have become a source of increased credit risk in the areas of corporate, housing, and consumer loans. Fernandez et al. indicated that the increased absenteeism of sick employees and the necessity of compulsory quarantine depleted human capital resources, especially in production and services provided directly to customers. As a consequence, the financial liquidity and ability to meet the credit obligations of these enterprises and their employees fell significantly. Such damage did not concern the IT industry, where the implementation of work in a remote system allowed for the continuity of operation and maintenance of current revenues [14]. The increase in credit risk, resulting from exposure to enterprises, was recorded in banks on both the cost and income side. The discontinuation of the servicing of loans increased banks' costs, i.e., the provisions for the potential realization of credit risk. On the other hand, the reduction in the number of potential borrowers and the opportunity for granting new loans significantly reduced banks' interest income, as well as other fees and commissions related to granting loans [25]. As a result, the banks weakened their ability to raise equity capital through retaining the net profit and to pay dividends to shareholders, which made it harder for them to attract new capital from the markets.

The banking sector stability during the COVID-19 pandemic was positively influenced by the actions of financial supervision authorities in most countries, which, in the form of regulations or verbal recommendations, i.e., the EU and the USA, obliged banks to suspend the payment of dividends for 2020 until the end of 2021 [26,27]. This policy contributed to the improvement of banks' capital ratios and an increase in their level of

security and stability. The correctness of these decisions is confirmed, inter alia, by the results of the study by Camilleri et al., who found that, in banks listed on stock exchanges in the Mediterranean countries operating in the years 2001–2016, the reduction in the dividend yield contributed to an increase in the market valuation of banks and an increase in their stability [28].

The increase in credit losses due to the pandemic, in many countries, has been exacerbated by risks from the state of public finances and the prevailing political situation. Tholl et al., when examining the yield spread between German and Italian government bonds found that political instability in Italy, as well as investors' fear of the country's exit from the euro area, contributed to the increase in the risk premium on Italian bonds and the increase in the costs for banks to obtain funds in the interbank markets, as well as significant volatility in the valuation of banks' assets [29]. The imposition of a sovereign risk premium by investors was also confirmed in the case of other large EU economies, i.e., France and Spain, whose bonds, especially after the 2008–2009 crisis, had elevated yields compared to the German bonds [30]. Moreover, Sensoy et al. noted that an increase in bond yields in crisis situations could be passed on through financial markets and lead to a deterioration in banks' performance and a weakening of their financial stability [31].

Public aid programs provided a real cushion for falling economy, although unequal across countries. IMF estimated that in EM countries, the public aid amounted to 5.5% of GDP and 20% of GDP in developed countries [32]. Additionally, Feyen et al., examining the financial sector policy response to the COVID-19 in 154 countries, stated that policy makers in richer and more populous countries have been significantly more responsive and have taken more policy measures. Moreover, countries with higher private debt levels tend to respond to the banking sector earlier with liquidity and funding measures [33].

The aid granted by governments and central banks took the form of direct aid, credit moratoria, and loan guarantees for enterprises taking out new loans. Such measures saved the banks against a sharp increase in the cost of credit risk, but on the other hand, limited the banks' ability to obtain interest income causing drop in banks' profitability. The credit moratoria prevented banks from charging any interests on the existing loans and the new-granted loans, secured by a state guarantee, provided lower spread on interest [25]. It is expected that this process, albeit in a milder form, may be long-term, which may exacerbate the banks' losses thus far.

The completion of a large number of public aid programs, scheduled for the first half of 2021, has become important for the financial situation of banks and requires in-depth analysis. According to the FSB, premature withdrawal of the State Treasury from public aid for enterprises may result in the collapse of many enterprises, especially micro and small ones, with lower security buffers and resources necessary to modify the current operating strategy. The negative impulse would translate into the situation of employees' households and an increase in credit risk at banks. On the other hand, the continuation of state aid may contribute to financial support for enterprises operating in sectors with poor development prospects and, at the same time, depletion of funds for sectors with good development prospects. It would also distort competition in the market and artificially retain low-performing entities in the market [12]. Similarly, the extension of the moratoria on loan repayment would adversely affect the long-term condition of banks, reducing their quality of loan monitoring and extending the period of maintaining increased write-offs for credit risk, in line with IFRS 9 [34].

Adopting a strategy of selective support for potential enterprises, even if it is economically justified, may nevertheless introduce a significant threat to the efficient functioning of the national economy. Although, in the long term, such a course of action may bring positive economic effects, in the short term, it may result in corporate bankruptcies, an increase in unemployment, and the necessity to incur additional costs of changing the sector of operation. This situation may contribute to an increase in the cost of credit risk, both in the segments of corporate and household loans [35]. The IMF research also shows that premature termination of supporting enterprises, in the process of fighting the effects

of COVID-19 and recovering to normality, may disturb the stability of a large number of banks in the EU [36].

The high level of risk still affects micro, small, and medium-sized enterprises as they do not have many capital buffers to survive the period of limited activity caused by the pandemic. For them, a bank loan is one of the basic sources of financing and maintaining financial liquidity. Providing these companies with loan guarantees would enable them to continue business operations, restore financial stability, and service existing loans on a regular basis [37]. Camous and Claeys also noted that, in the conditions of the financial crisis resulting from the COVID pandemic, the fact of participation in the integrated banking system of the euro area was of significant importance for the results of banks. The monetary and fiscal aid policy in the euro area did not always support enterprises and banks in individual countries in the same way. However, the creation of the fund to fight coronavirus crisis will allow companies to have equal chances of emerging from the crisis situation, especially in the regions that have been most severely affected by the pandemic and, in consequence, improve the situation of banks that are lending to these enterprises [38].

Based on this literature background, the following hypotheses were formulated:

**Hypothesis 1.** *Equity of banks in CESE countries are able to absorb relatively high levels of credit losses without violating regulatory capital requirements.*

**Hypothesis 2.** *Smaller banks are able to absorb larger increases in nonperforming loans than in the case of larger entities, while maintaining regulatory capital requirements.*

## 3. Materials and Methods

The study used a scenario-based model developed by Barua and Barua [39] to assess the implications of the COVID-19 pandemic for the profitability and financial stability of banks in CESE countries. In short, the test consists of imposing nonperforming loans shocks to the balance sheet and P&Ls positions and checking the resulting value of the bank's equity and capital ratios. Alternatively, the stability of enterprises can be tested using linear programming and the probability of default method [40–42], but they require, inter alia, market data not available to all banks included in the sample. The research is based on data at the end of 2020, and the model examines the financial situation of banks at the end of next year. The capital ratios, TCR and CET1, measure the stability and ROA profitability of a bank. The shock to the banking sector results from the extraordinary growth of NPLs, which reduces the equity of banks and their interest income on lost loans.

The shock caused by the COVID-19 pandemic was most manifested in the form of credit risk resulting from a significant deterioration in the creditworthiness of households and enterprises operating in almost all sectors of the economy and, in consequence, in line with IFRS-9, led to an increase in credit risk charges [43]. The pandemic was also a source of other risks that emerge in banking activities. Restrictions on customer access to bank branches, the inability to fill all workplaces due to the sickness of employees, their family members, and the quarantine obligation, the accumulation of tasks attributable to the reduced number of bank staff were the source of operational risk. On the other hand, sharp drops in the valuation of equity instruments, especially bonds, and changes in exchange rates have significantly increased the level of market risk in banks. However, the scale of these types of risk was much smaller compared to the credit risk. Therefore, due to the lack of sufficient data necessary to estimate the losses incurred in connection with the implementation of operational or market risks and the fact that the scale of the impact of these types of risk was much smaller than the credit risk, in the examination procedure, the effects of COVID-19 shock were considered as the only cause of the increase in the nonperforming loans.

The model used the following items of the balance sheet and income statement: total assets (*TA*), gross loans (*GL*), net loans (*NL*), nonperforming loans (*NPL*), equity represented

by Total Capital (*TC*), Tier 1 Capital (*T1*), risk weighted assets (*RWA*), and net income (*NI*). The interest rate on loans is represented by the interest rate of the three-month interbank rate in each country ($r_c$). It is assumed that, evenly throughout the year, in accordance with IFRS-9, banks record loan impairment at the moment of recognizing the risk of loan default and create provisions for losses for this reason. As a result, it reduces the value of properly serviced loans and equity of the bank. Based on these assumptions, the following values can be determined for the bank i:

The increase in the value of non-performing loans at the end of the year $\Delta NPL_i$ resulting from the deterioration by $\delta$ the net loans serviced at the beginning of the year $NL_i$ ($\delta$ ranges from 1% to 20%):

$$\Delta NPL_i = NL_i \cdot \delta \tag{1}$$

The residual value of the bank's equity (*TC* and *T1*) resulting from the reduction in initial equity by the value of newly recognized nonperforming loans $\Delta NPL_i$ is:

$$TCr_i = TC_i - \Delta NPL_i; \qquad T1r_i = T1_i - \Delta NPL_i \tag{2}$$

Assuming that the bank's financial assets consist mainly of loans and bonds issued by the treasury or a central bank with a zero-risk weight, the value of risk-weighted assets $RWA_i$ depends mainly on the value of loans $NL_i$. The average value of the risk weight $w_i$ for the bank's loan portfolio is:

$$w_i = \frac{RWA_i}{NL_i} \tag{3}$$

The residual value of risk-weighted assets at the end of the year equals to:

$$RWAr_i = RWA_i - w_i \cdot \Delta NPL_i \tag{4}$$

The residual value of capital ratios ($TCRr_i$, $CET1r_i$) equal to:

$$TCRr_i = \frac{TCr_i}{RWAr_i}; \quad CET1r_i = \frac{T1r_i}{RWAr_i} \tag{5}$$

Assuming a gradual and steady increase in the value of nonperforming loans throughout the year and the average interest rate on these loans in a given country $r_c$, the final net income equals:

$$NIr_i = NI_i - 0.5 \cdot r_c \cdot \Delta NPL_i \tag{6}$$

The final value of the return on assets at the end of the year equals to:

$$ROAr_i = \frac{NIr_i}{TA_i - \Delta NPL_i} \tag{7}$$

The study covered 141 banks operating in 18 CESE countries, i.e., Bosnia and Herzegovina (BA–4 banks), Bulgaria (BG–6), Belarus (BY–4), Chechia (CZ–11), Estonia (EE–5), Croatia (HR–9), Hungary (HU–8), Lithuania (LT–3), Latvia (LV–4), Moldova (MD–3), Montenegro (ME–3), North Macedonia (MK–3), Poland (PL–12), Romania (RO–10), Serbia (RS–5), Russia (RU–32), Slovenia (SI–4), Slovakia (SK–5), Ukraine (UA–10). The sample includes 74 banks listed on the stock exchange. The data on individual banks are provided by S&P Global database. Table 1 shows the descriptive statistics of the analysed annual data at the end of 2020. The study used scenarios in which the value of nonperforming loans increases by $\delta$ percent in the following year. Taking into account the fact that in the analysed sample of banks the maximum value of the NPL ratio was 23.7%, it was assumed that the $\delta$ ratio will be in the range from 1% to 20%. When calculating the value of the lost interest income, the interest rate for three-month loans on the interbank market in individual countries at the end of 2020 was used. This type of rate is commonly used to calculate interest on loans. The rates were taken from the websites of national banks.

**Table 1.** Descriptive statistics for the variables applied (billions USD).

| Variable | Number of Observations | Average | Standard Deviation | Quartile 1 | Median | Quartile 3 |
|---|---|---|---|---|---|---|
| *TA* | 141 | 18.20 | 46.80 | 1.79 | 6.02 | 19.13 |
| *RWA* | 141 | 14.92 | 47.36 | 1.64 | 4.81 | 11.26 |
| *TC* | 141 | 1.97 | 6.04 | 0.31 | 0.87 | 2.53 |
| *T1* | 141 | 1.82 | 5.75 | 0.26 | 0.83 | 2.24 |
| *NL* | 141 | 10.47 | 30.31 | 1.10 | 3.18 | 10.65 |
| *NI* | 141 | 0.19 | 0.88 | 0.01 | 0.04 | 0.15 |
| *NPL* (%) | 141 | 8.12 | 10.80 | 2.67 | 5.35 | 8.40 |
| *TD/TA* (%) | 141 | 63.41 | 28.03 | 68.83 | 75.12 | 81.13 |
| *GL/TA* (%) | 141 | 58.72 | 15.79 | 48.72 | 59.34 | 69.86 |
| *C/I* (%) | 141 | 58.75 | 18.08 | 50.06 | 57.77 | 70.64 |
| *C/TA* (%) | 141 | 2.61 | 1.53 | 1.61 | 2.14 | 3.15 |
| *NIM* (%) | 141 | 2.90 | 1.90 | 1.93 | 2.53 | 3.25 |
| *ROE* (%) | 141 | 7.66 | 11.95 | 3.49 | 6.63 | 10.99 |
| *ROA* (%) | 141 | 0.84 | 1.21 | 0.36 | 0.75 | 1.20 |
| $r_c$ (%) | 18 | 5.33 | 3.12 | 2.64 | 5.00 | 6.10 |

Note: TA—total assets, RWA—risk weighted assets, TC—total capital, T1—Tier 1 capital, NPL—nonperforming loans, NI—net income, TD—total deposits, GL—gross loans, C—operating costs, NIM—net interest income, ROE—return on equity, ROA—return on assets, $r_c$—interest rate in the country.

It is worth noting that the average values of the balance sheet variables characterizing banks are much larger than their median, which indicates that in the sample, the group of smaller banks is much more numerous than the large ones. In terms of the quality of the loan portfolios, in nearly 75% of banks, the share of nonperforming loans is lower than the average value for the sample. On the other hand, the fact that a much larger proportion of banks have higher than the sample's average shares of deposits and loans in the balance sheet total indicates that the financial intermediation model is dominant in the operations of banks in the CESE region. Such a business strategy may support banks' cost efficiency as, in most banks, the ratio of operating costs to banking income and to total assets is lower than the sample averages. However, better cost efficiency does not appear to be a sufficient factor to improve the profitability of banks. In most banks, the net interest margin (NIM), ROE, and ROA ratios are lower than their sample mean values. These features indicate that, in the CESE countries, there are mostly smaller banks that implement traditional banking, with higher cost efficiency but lower profitability. Interest rates on the interbank market were relatively evenly distributed in the analysed sample.

The stress-test model was applied to the entire group of banks operated in CESE countries. In addition, in order to assess the sensitivity of the equity capital to the extraordinary increase in nonperforming loans caused by the COVOD-19 pandemic, the following categories were distinguished depending on the location and characteristics of banks:

- EU–banks operating in EU countries;
- Non-EU–banks operating in countries outside the EU;
- EU-Euro–banks operating in the euro area countries;
- EU-Non-Euro–banks operating in countries outside the euro area;
- Large–banks with total assets greater than the median sample;
- Small–banks with assets lower than the median sample;
- Over-Cap–banks with equity higher than the median sample;
- Under-Cap–banks with equity lower than the median sample;
- Public–banks listed on the stock exchange;
- Non-Public–banks not listed on the stock exchange.

## 4. Results of Modelling and Discussion

In the first stage, the impact of a δ percent increase in nonperforming loans on the bank's equity was examined. The calculations were conducted for the value of δ changing from 1% to 20% every 1 pp. Using the Formula (1), the values of nonperforming loans

were determined as a result of the one-year exposure to the COVID-19 pandemic and then, the final values of equity (Formula (2)). Subsequently, on the basis of the Formula (3), the values of credit risk weights $w$ of the portfolios were estimated. The average value of the $w$ ratio, for the banks included in the sample, was 49.8% and is comparable to the average value of the risk weight ratio for exposures to non-financial corporations reported by the banks notified at the EBA as of the end of June 2020, amounting to 54% [44]. Based on Formulas (4) and (5), the final values of risk-weighted assets and capital ratios (TCR and CET1) were determined, respectively. Table 2 presents the values of $\delta_r$ for which the average TCR and CET1 ratios in each country would drop below 8% and 6%, respectively.

**Table 2.** Projected values of the $\delta_r$ index for TCR and CET1 ratios below required levels (%).

| Country | BA | BG | BY | CZ | EE | HR | HU | LT | LV | MD |
|---|---|---|---|---|---|---|---|---|---|---|
| $\delta$ for TCR < 8% | 11 | 9 | 7 | 13 | 14 | 15 | 15 | 11 | 16 | 11 |
| $\delta$ for CET1 < 6% | 11 | 10 | 9 | 14 | 15 | 15 | 15 | 14 | 16 | 11 |
| **Country** | **ME** | **MK** | **PL** | **RO** | **RS** | **RU** | **SI** | **SK** | **UA** | **CESE** |
| $\delta$ for TCR < 8% | 14 | 14 | 11 | 17 | 15 | 16 | 17 | 8 | 14 | 12 |
| $\delta$ for CET1 < 6% | 13 | 12 | 11 | 17 | 15 | 13 | 16 | 8 | 14 | 12 |

Note: BA—Bosnia and Herzegovina, BG—Bulgaria, BY—Belarus, CZ—Czechia, EE—Estonia, HR—Croatia, HU—Hungary, LT—Lithuania, LV—Latvia, MD—Moldova, ME—Montenegro, MK—North Macedonia, PL—Poland, RO—Romania, RS—Serbia, RU—Russia, SI—Slovenia, SK—Slovakia, UA—Ukraine, CESE—Central Eastern South Europe.

The results of the calculations indicate that, at the end of 2020, the capital position of banks in the CESE region was relatively strong. During the following year, the banking sector had enough equity capital to absorb a 12% increase in nonperforming loans due to the COVID-19 pandemic. This condition applies to both the requirement to maintain an appropriate level of total capital and Tier 1 capital. Obtained results indicate that banks operating in the CESE region are relatively resilient to the shocks caused by the extraordinary increase in nonperforming loans caused by the COVID-19 pandemic. They are also consistent with the results of the IMF research [35] on European banks, which stated that, despite significant drops in capital ratios in 2020, banks were still immune to shocks. Such a situation was confirmed in the banking sectors of the euro area and other CESE countries, including Czechia, Hungary, Poland, Romania, and Ukraine, where good capital equipment allowed them to maintain the stable situation of the banking sector in the first quarters of the pandemic in 2020 [45–49].

However, the sensitivity of banks' capitals to extraordinary losses varies across countries of the region. Differentiation occurs both among countries with a similar geographic location as well as those belonging to the EU or the euro area. For example, among Southern European countries, banks' resilience in the Republic of Serbia (15%) is much higher than in Bosnia and Herzegovina (11%). In the Euro area, banks in Latvia and Slovenia are able to absorb a 16% increase in nonperforming loans, compared to only 8% in Slovakia. On the other hand, in the EU countries not belonging to the Euro area, banks in Romania can absorb credit losses increased by 17%, while only by 9% in Bulgaria.

In the entire sample, the lowest resilience against COVID-19 credit losses occurs in Belarus, Slovakia, and Bulgaria, where banks can withstand an increase in nonperforming loans by 8–9%. In turn, banks in Romania, Slovenia, and Latvia had the highest capital resilience to extraordinary losses. In their cases, a breach of one of the capital requirements would only take place after the increase in nonperforming loans by 16%. These results are consistent with the results of the IMF, indicating that the level of resistance of European banks to shocks is highly diversified among countries [36].

Figures 1 and 2 present graphs of the dependence between the level of capital ratios, TCR and CET1, respectively, on the value of the growth rate of nonperforming loans caused by the COVID-19 pandemic.

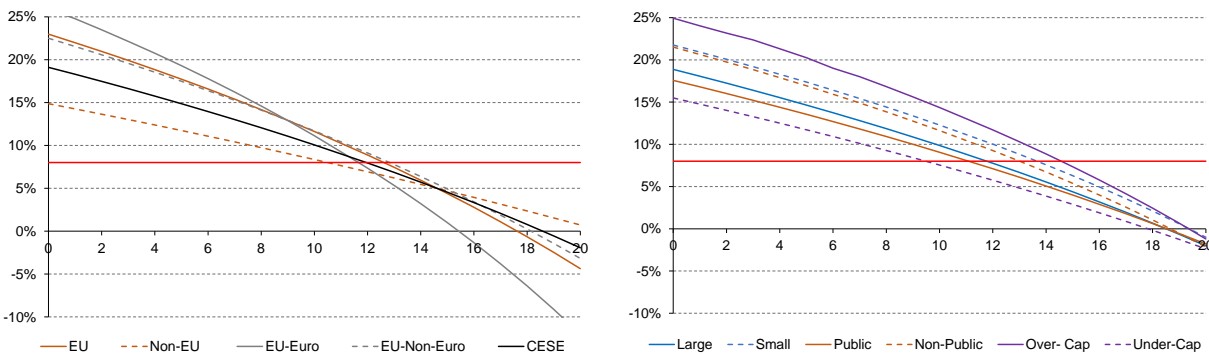

**Figure 1.** Changes in TCR depending on the value of δ indicator.

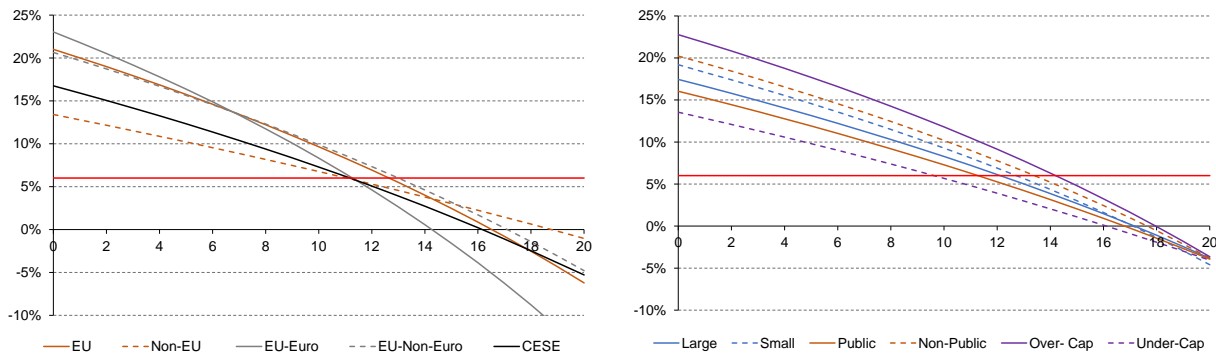

**Figure 2.** Changes in CET1 depending on the value of δ indicator.

The calculation results indicate that the initial capital position of banks located in EU countries was better than that of non-EU countries. This allows for the maintenance of capital requirements with an increase in nonperforming loans by 13%, compared to 11% in non-EU countries. However, the pace of deterioration of equity capital, along with the increase in the scale of credit losses, in the EU banks was higher than in countries outside the EU, which means that, with the further deterioration of the loans' quality, the capital position would reverse on the benefit of non-EU banks. A similar phenomenon occurs in the case of banks located in and outside the Euro area. Although the capital ratios of the first group were initially higher, the much stronger pace of deterioration of capital position, depending on the increase in credit losses, means that the capital requirements, TCR and CET1, of the Euro area banks are violated, respectively, for the δ of 12% and 11%, while for banks located in other EU countries, both for TCR and CET1, for δ equal to 13%.

A comparison of the sensitivity of capital ratios at banks, differentiated by the value of total assets, indicates that smaller entities have a better capital position and maintain this advantage also in the event of large increases in the nonperforming loans. Smaller banks do not meet the capital requirements for TCR and CET1 ratios, in the case of an increase in nonperforming loans by 14% and 13%, respectively, while larger banks—in both cases, increase by 12%. These results are in line with the results of Flogel and Gartner, who found that, in Germany, smaller regional banks more effectively responded to the deterioration of economic conditions during the global financial crisis of 2007–2009. They granted more loans to local enterprises, compared to the group of the largest German banks, which allowed them to maintain profitability and an appropriate value of equity [50]. The better resilience against an increase in nonperforming loans would also be explained, partly, with the fact that some smaller banks did not fully report losses associated with COVID-19. Such behaviour of banks confirms the research of Neef and Schandlbauer, who found that, during the crisis, smaller banks and less capitalized banks in Europe extended more new loans, inter alia, to support their weaker borrowers and avoid loan loss recognition and creation of write-offs reducing their equity [51]. The analysis of the banks with stronger

and weaker capital standing showed that the over-cap group was the most resilient to the extraordinary increase in credit losses caused by the COVID-19 pandemic and was the best performer of any group analysed so far. The critical credit loss indicator, for both TCR and CET1, equalled to 15%, while the under-cap group equalled 10%. The results are consistent with the results obtained by the IMF, which show that the quality of a bank's response to shocks caused by a pandemic depends on the bank's initial capital position and its overall financial situation [1].

A similar relationship exists between groups of the public banks and the non-public banks. The first ones are obligated to apply more restrictive regulations, referring to non-performing loans set by the IFRS-9, and are regularly audited by external institutions [34]. The non-public banks, however, are able to postpone the date of recognizing the risk of defaulting loans and recording of nonperforming loans, which may explain their better capital position after the first quarters of the COVID-19 pandemic. As a result in their case, the capital requirements for TCR and CET1 are no longer met with an increase in non-performing loans by 13%, while they increase by 11% in the case of public banks.

Another area of research was the assessment of the impact of the COVID-19 credit losses on the profitability of banks. As in the case of capital requirements, the results of the analysis obtained for individual banks are aggregated at the level of countries, groups of countries, and groups of banks with common characteristics. The relationships between the ROA level and the increase in loan losses for individual groups of banks are presented in Figure 3.

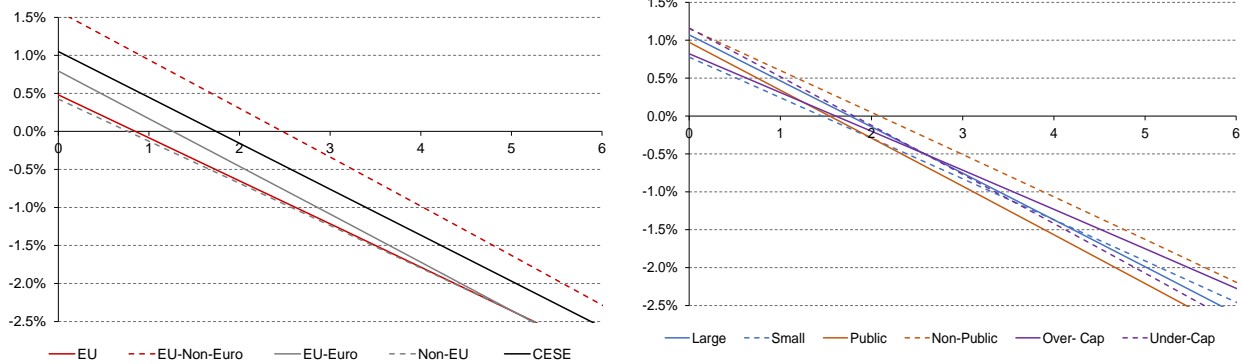

**Figure 3.** Changes in ROA depending on the value of δ indicator.

The results of the estimates indicate that the profitability of banks is highly sensitive to the extraordinary losses incurred by banks during the COVID-19 pandemic. In the case of banks operating in most countries, an increase in nonperforming loans by 2% caused the net profit, and thus the profitability of assets, to become negative. Banks in non-EU countries were able to maintain positive profitability with higher credit losses than the EU banks. In the case of banks operating in the EU, the entities located in the Euro area maintained positive profitability with higher losses than those from other EU countries. In the case of the group of large banks, their advantage over the small entities may come from the usage of the economies of scale in their operations and reducing operating costs in such a way. The results obtained are in line with the results of Asimakopoulos et al., who indicates that, in a period of crisis, entities characterized by high asset values, annual sales, or investments made, are able to maintain a positive profitability of their activities more easily compared to smaller competitors. On the other hand, operating in the EU was not conducive to maintaining profitability during the crisis [52]. The results are also in line with the results of Dadoukis et al., who indicates that larger banks and those who were better IT equipped in the pre-pandemic period during the first quarters of the pandemic, better maintained their market position, profitability, and financial stability [53]. Additionally, larger banks have more capacity to introduce non-interest services for diversification of income and improvement of profitability [54–57].

## 5. Conclusions

The impact of COVID-19 on the state of the economy and financial situation of enterprises varies and is stronger in the case of the country and region with the income structure focused on tourism and services directly delivered to the customer, micro, small, and medium-sized enterprises owned and operated by a minority (including women).

The reduction in the scale of the COVID-19 pandemic and the gradual increase in economic activity do not mean that the risk of extraordinary losses on nonperforming loans will disappear. The rise in the nonperforming loans may come from the increase in morbidity, the re-imposition of restrictive sanitary requirements, restrictions imposed on economic activity, as well as too early or too late termination of state aid programs. It should be considered that, in times of crisis, shocks in nonperforming loans may be accompanied by a run on banks and liquidity disruptions [58]. Leoni [59] and Lagoarde-Segot and Leoni [60] refer to crises caused by the spread of the HIV epidemic and the collapse of the microfinance institutions and banks lending to the poor in EM countries, which caused not only a default on loan obligations but also excessive demand for cash to ensure daily payments. A similar situation may arise when the COVID-19 pandemic develops in countries with less developed health systems.

The conducted research on the sensitivity of the capital position (TCR and CET1) of banks operating in 18 CESE countries to the growth of NPLs indicates that the first hypothesis was confirmed. At the end of 2020, in the CESE region, banks had relatively strong capital equipment, allowing them to maintain capital requirements with a 12% increase in nonperforming loans. However, the level of banks' resilience to the extraordinary shocks, caused by COVID-19, varied across the region. Equity capital of banks located in EU countries is less sensitive to extraordinary increases in nonperforming loans than their competitors in non-EU countries, which include results from their better capital endowment to date, as well as lower levels of nonperforming loans.

Smaller entities, usually unlisted on the stock exchange, maintain regulatory capital parameters with larger increases in nonperforming loans than their larger and listed competitors, which confirms the second hypothesis. However, there is a risk that these groups of banks postpone the creation of provisions for nonperforming loans, in order to improve their financial statements, and also increase new lending to lower the NPL ratio. Such a situation may be a source of additional credit risk as well as a sharp deterioration of the capital position of banks in the future.

The profitability of banks in CESE countries is relatively sensitive to the appearance of increased amounts of nonperforming loans. Their increase by several percent is the source of the negative profitability of banks. Larger banks better preserve their positive profitability, among other, due to benefits of economy of scale. Banks operating in countries outside the EU, and not listed on stock exchanges, have a greater ability to remain profitable when there is a sharp deterioration in the quality of the loan portfolio.

The results of the performed examination of banks' resilience to extraordinary credit losses indicate, to financial supervisors and to banking market analysts, the assessment of how strongly the increase in credit losses is able to withstand the equity of banks in CESE, as well as in individual countries of the region. The conclusions of the research also indicate which banks are most vulnerable to the worsening of the COVID-19 pandemic situation and require enhanced supervision.

**Funding:** This research received no external funding.

**Institutional Review Board Statement:** Not applicable.

**Informed Consent Statement:** Not applicable.

**Data Availability Statement:** The data presented in this study are available on request.

**Conflicts of Interest:** The author declares no conflict of interest.

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
