# Peer review of "The Impact of COVID-19 on Bank Equity and Performance: The Case of Central Eastern South European Countries"

_sustainability, doi:10.3390/su131911036_

Round 1

Reviewer 1 Report

I suggest to better specify the results and the methodolody

Author Response

Response to Reviewer 1 Comments

Comments

Responses

I suggest to better specify the results and the methodology

I would like to sincerely thank you for your valuable suggestions and remarks that allowed me to modify and improve quality of my article.

I rewrote the section Materials and Methodology. Additionally I included the idea of the methodology to Abstract.

Concerning results, I extended the discussion of the results by referring them simultaneously to the results presented in the new literature items added.

I significantly expanded the scope of the literature, using new items to present the economic background as well as to discuss the results and conclusions of the study.

Reviewer 2 Report

This interesting paper examines Covid-19 and the banking sector in the CESE countries. 

All calculations seem to be correct. This is of course important. But there is room for improvement.

Most importantly, the literature review should be improved in order to show how this paper can help to improve our understanding of the issues that are of relevance in the field of financial economics that is examined here. Clearly, Tholl et al. (2020) should be cited. This paper examines funding issues in the Italian banking industry and also discusses the implications of the Covid-19 crisis. In this context Tholl et al. (2020) have highlighted the importance of sovereign credit risk. Therefore, it could be a good idea to also briefly examine the European government debt crisis – and to cite two relevant papers (e.g., Sibbertsen, Wegener and Basse, 2014 and Sensoy et al., 2019).

Citing some additional papers that examine the consequences of Covid-19 for the banking industry in specific European countries might also be helpful to imporce the literature review (e.g., Camous and Claeys, 2020 and Flögel and Gärtner, 2020). One paper that analyzes the global dimension of this problem could also be cited (e.g., Wu and Olson, 2020).

The paper already discusses the important study by Simoens and Vander Vennet (2021) who analyze “shock absorbers” in banks and the Covid-19 crisis. This is good! They discuss the cost efficiency of banks, liquidity buffers and capital buffers. Given that that the paper focuses on the capital position of banks it might probably also be a good idea to briefly review the literature that examines the dividend policy of banks in Europe after the sovereign debt crisis. There are a number of relevant papers that could (and should) be cited in this context. In fact, Dombret, Gündüz and Rocholl (2019) have also done this in their paper that is examining a similar topic. In fact, it might be a good idea to discuss dividend policy issues in the banking industry in some detail because baking regulators have started to look at this point in more detail. Doing this will certainly improve the paper and would help to make this study more important in its field because dividend policy issues are highly relevant in financial economics and corporate finance. I would suggest to provide a comprehensive overview of dividend policy issues in the European financial services industry emphasizing the consequences for the capital position of banks. In addition, regulatory issues could also be briefly discussed in this context. Given the focus of this paper it might for example be a good idea cite Camilleri, Grima and Grima (2019) who examine Mediterranean banks and their dividend policies. There is quite a lot of additional literature that could be cited here in order to improve the quality of the literature review.

Moreover, it would be a good idea to discuss some ideas for future research in the conclusion. Goodell (2020) could be helpful at this point!

Given the role of Greece in the European government debt crisis it could also be helpful to cite Asimakopoulos, Samitas, and Papadogonas (2009) – they have examined firm-specific and economy wide determinants of firm profitability in the Greek economy.

Moreover, this paper focuses quite strongly on SMEs. Thus, it might be a good idea to cite and briefly discuss Juergensen, Guimón and Narula (2020).

Additional literature:

Asimakopoulos, I., Samitas, A., & Papadogonas, T. (2009). Firm‐specific and economy wide determinants of firm profitability: Greek evidence using panel data. Managerial Finance, 35, 930-939.

Camilleri, S. J., Grima, L., & Grima, S. (2019). The effect of dividend policy on share price volatility: an analysis of Mediterranean banks’ stocks. Managerial Finance, 45, 348-364.

Dombret, A., Gündüz, Y., & Rocholl, J. (2019). Will German banks earn their cost of capital?. Contemporary Economic Policy, 37, 156-169.

Camous, A., & Claeys, G. (2020). The evolution of European economic institutions during the COVID‐19 crisis. European Policy Analysis, 6, 328-341.

Flögel, F., & Gärtner, S. (2020). The COVID‐19 Pandemic and Relationship Banking in Germany: Will Regional Banks Cushion an Economic Decline or is A Banking Crisis Looming?. Tijdschrift voor economische en sociale geografie, 111(3), 416-433.

Goodell, J. W. (2020). COVID-19 and finance: Agendas for future research. Finance Research Letters, 35, 101512.

Juergensen, J., Guimón, J., & Narula, R. (2020). European SMEs amidst the COVID-19 crisis: assessing impact and policy responses. Journal of Industrial and Business Economics, 47(3), 499-510.

Sensoy, A., Nguyen, D. K., Rostom, A., & Hacihasanoglu, E. (2019). Dynamic integration and network structure of the EMU sovereign bond markets. Annals of Operations Research, 281, 297-314.

Sibbertsen, P., Wegener, C., & Basse, T. (2014). Testing for a break in the persistence in yield spreads of EMU government bonds. Journal of Banking and Finance, 41, 109-118.

Tholl, J., Schwarzbach, C., Pittalis, S., & von Mettenheim, H. J. (2020). Bank funding and the recent political development in Italy: What about redenomination risk?. International Review of Law and Economics, 64, 105932.

Wu, D. D., & Olson, D. L. (2020). The effect of COVID-19 on the banking sector. In Pandemic Risk Management in Operations and Finance (pp. 89-99). Springer, Cham.

Author Response

Response to Reviewer 2 Comments

Comments

Responses

This interesting paper examines Covid-19 and the banking sector in the CESE countries. All calculations seem to be correct. This is of course important. But there is room for improvement.

I would like to sincerely thank you for your valuable remarks, as well as suggestions of new literature items, that allowed me to modify and improve quality of my article. As a result I significantly extended the scope of the literature, using new items to present the economic background as well as to discuss the results and conclusions of the study.

Most importantly, the literature review should be improved in order to show how this paper can help to improve our understanding of the issues that are of relevance in the field of financial economics that is examined here. Clearly, Tholl et al. (2020) should be cited. This paper examines funding issues in the Italian banking industry and also discusses the implications of the Covid-19 crisis. In this context Tholl et al. (2020) have highlighted the importance of sovereign credit risk. Therefore, it could be a good idea to also briefly examine the European government debt crisis – and to cite two relevant papers (e.g., Sibbertsen, Wegener and Basse, 2014 and Sensoy et al., 2019).

I added the proposed items and included in the text the issues of risk, which may additionally increase banks' losses.

Citing some additional papers that examine the consequences of Covid-19 for the banking industry in specific European countries might also be helpful to imporce the literature review (e.g., Camous and Claeys, 2020 and Flögel and Gärtner, 2020). One paper that analyzes the global dimension of this problem could also be cited (e.g., Wu and Olson, 2020).

I added the proposed items and included in the text the issues related to the importance of the integration of the economy and financial markets during the crisis.

The paper already discusses the important study by Simoens and Vander Vennet (2021) who analyze “shock absorbers” in banks and the Covid-19 crisis. This is good! They discuss the cost efficiency of banks, liquidity buffers and capital buffers. Given that that the paper focuses on the capital position of banks it might probably also be a good idea to briefly review the literature that examines the dividend policy of banks in Europe after the sovereign debt crisis. There are a number of relevant papers that could (and should) be cited in this context. In fact, Dombret, Gündüz and Rocholl (2019) have also done this in their paper that is examining a similar topic. In fact, it might be a good idea to discuss dividend policy issues in the banking industry in some detail because baking regulators have started to look at this point in more detail. Doing this will certainly improve the paper and would help to make this study more important in its field because dividend policy issues are highly relevant in financial economics and corporate finance. I would suggest to provide a comprehensive overview of dividend policy issues in the European financial services industry emphasizing the consequences for the capital position of banks. In addition, regulatory issues could also be briefly discussed in this context. Given the focus of this paper it might for example be a good idea cite Camilleri, Grima and Grima (2019) who examine Mediterranean banks and their dividend policies. There is quite a lot of additional literature that could be cited here in order to improve the quality of the literature review.

I added the proposed items and included in the text the issues related to the importance of the dividend policy pursued by banks, as well as financial supervision for the results and financial stability of banks.

Moreover, it would be a good idea to discuss some ideas for future research in the conclusion. Goodell (2020) could be helpful at this point!

I included the proposed item and included it in the Conclusions, pointing out that the crisis resulting from COVID-19 may be further aggravated by the violent and simultaneous behavior of bank customers.

Given the role of Greece in the European government debt crisis it could also be helpful to cite Asimakopoulos, Samitas, and Papadogonas (2009) – they have examined firm-specific and economy wide determinants of firm profitability in the Greek economy.

I added the proposed item and included the discussion of results as an example of research on the importance of the bank's size, as well as the location of its operation on the its performance.

Moreover, this paper focuses quite strongly on SMEs. Thus, it might be a good idea to cite and briefly discuss Juergensen, Guimón and Narula (2020).

I added the proposed item and included in the discussion on the impact of the size of the enterprise on the level of risk in its operation and the size of its results.

Reviewer 3 Report

Thanks for the efforts by authors about the effect of COVID19 on bank equity and performance. 

I have serious concerns about the paper:

  1. Structure: The paper does not address hypothesis and theoretical background. It just lists out some prior studies and jump right into the methods. Even if you get the results there is nothing that support your results. I think this should be heavily revised.
  2. Data: you must address the data sources and descriptive information about the data. I don't really get why you chose those countries and what kind of contribution from those countries and banks. Also please present some descriptive statistics about the sample like size, debt ratio, performance, other control variables. 
  3. Format: The tables and Figures are wrongly formatted and especially the reference is not acceptable (format). 

Author Response

Response to Reviewer 3 Comments

Comments

Responses

Thanks for the efforts by authors about the effect of COVID19 on bank equity and performance. 

I have serious concerns about the paper.

I would like to sincerely thank you for your valuable remarks and suggestions, that allowed me to modify and improve quality of my article.

I significantly extended the scope of the literature, using new items to present the economic background as well as to discuss the results and conclusions of the study.

Structure: The paper does not address hypothesis and theoretical background. It just lists out some prior studies and jump right into the methods. Even if you get the results there is nothing that support your results. I think this should be heavily revised.

I restructured the article structure. In Introduction, I stated research hypotheses, as well as extended the analysis of the results. In Introduction and Conclusions I indicated possible applications of the results of my research.

Data: you must address the data sources and descriptive information about the data. I don't really get why you chose those countries and what kind of contribution from those countries and banks. Also please present some descriptive statistics about the sample like size, debt ratio, performance, other control variables.

I extended the scope of the presented data on banks and modified the justification for the selection of the analyzed countries. In addition, I extended the characteristics of the banks in the sample, also taking into account the data items added.

Format: The tables and Figures are wrongly formatted and especially the reference is not acceptable (format).

I corrected the formatting of tables and charts, as well as the way of writing the references.

Reviewer 4 Report

Review for

The Impact of COVID-19 on Bank Equity and Performance.The Case of Central Eastern South European Countries

The level of originality of the paper is high. The literature review and proposed methodology are properly discussed and compared to the previous studies.

  • Abstract is very weak. It needs to rewrite at all in accordance with journal rules.
  • The introduction section has benefit from having a clearer structure of what to expect in the paper. Furthermore, the author(s) would benefit from being more concise in their writing, as much of the content was redundant and overemphasized. While it is good practice to assume the reader has no prior knowledge of the content, a topic and/or discussion does not need to be explained over and over again if it is stated both adequately and appropriately once.
  • In this paper, authors used only 32 sources, containing both historical and fundamental works, as well as the latest scientific research on this topic. But the literature review can be structured. The papers discussed many points of this study. Please, discuss these papers:

An, J., Mikhaylov, A., Jung, S.-U. (2021). A Linear Programming Approach for Robust Network Revenue Management in the Airline Industry. Journal of Air Transport Management, 91(3), 101979. https://doi.org/10.1016/j.jairtraman.2020.101979

An, J., Mikhaylov, A., Richter, U.H. (2020) Trade War Effects: Evidence from Sectors of Energy and Resources in Africa. Heliyon, 6, e05693. https://doi.org/10.1016/j.heliyon.2020.e05693

Mishina V.Yu., Khomyakova l.I. Dedollarization and settlements in national currencies: Eurasian and Latin American experience. Voprosy Ekonomiki. 2020;(9):61-79. https://doi.org/10.32609/0042-8736-2020-9-61-79

Some conclusions contribute to the study of the problem. The author does not formulate the problem itself – it makes impossible to analyse the contribution of the paper. The aim or the question of the paper (or even the hypothesis of the author) are formulated.

  • Overall, it is very clear to grasp understanding of the manuscript and content in its current state. I strongly advise using hypothesis points to articulate and/or express material in scientific writing. Publication of this piece seems likely in any reputable scientific periodical after a correction in the writing of the manuscript.
  • Table 1-2 is important to explore the specifics. Author needs to wide it. Some conclusions can contribute to the study of the problem.
  • Author needs to add more details on the range of simulation considered in this work should be clearly outlined within the abstract. The current statements are vague and too general to get an idea of the work that have been accomplished.

The paper possesses a proper form of well-structured and readable technical language of the field and represents the expected knowledge of the journal`s readership. There are minor errors in English, but this does not affect the general nature of the work. The current study brings many new to the existing literature or field. For one, the author(s) seem to have a good grasp of the current literature on their topic area (i.e., recent literature and seminal texts relevant to their study is not cited/referenced).

Author Response

Response to Reviewer 4 Comments

Comments

Responses

The level of originality of the paper is high. The literature review and proposed methodology are properly discussed and compared to the previous studies.

I would like to sincerely thank you for your valuable remarks, as well as suggestions of new literature items, that allowed me to modify and improve quality of my article.

Abstract is very weak. It needs to rewrite at all in accordance with journal rules.

I rewrote Abstract and adjusted it to the rules of the journal.

The introduction section has benefit from having a clearer structure of what to expect in the paper. Furthermore, the author(s) would benefit from being more concise in their writing, as much of the content was redundant and overemphasized. While it is good practice to assume the reader has no prior knowledge of the content, a topic and/or discussion does not need to be explained over and over again if it is stated both adequately and appropriately once.

I restructured the Introduction and removed duplicate and redundant descriptions.

In this paper, authors used only 32 sources, containing both historical and fundamental works, as well as the latest scientific research on this topic. But the literature review can be structured. The papers discussed many points of this study. Please, discuss these papers:

An, J., Mikhaylov, A., Jung, S.-U. (2021). A Linear Programming Approach for Robust Network Revenue Management in the Airline Industry. Journal of Air Transport Management, 91(3), 101979. https://doi.org/10.1016/j.jairtraman.2020.101979

An, J., Mikhaylov, A., Richter, U.H. (2020) Trade War Effects: Evidence from Sectors of Energy and Resources in Africa. Heliyon, 6, e05693. https://doi.org/10.1016/j.heliyon.2020.e05693

Mishina V.Yu., Khomyakova l.I. Dedollarization and settlements in national currencies: Eurasian and Latin American experience. Voprosy Ekonomiki. 2020;(9):61-79. https://doi.org/10.32609/0042-8736-2020-9-61-79

I extended the literature review to include new aspects of risk emerging in banks as a result of COVID-19.

In addition, I added the proposed literature items in the area of research methodology.

Some conclusions contribute to the study of the problem. The author does not formulate the problem itself – it makes impossible to analyse the contribution of the paper. The aim or the question of the paper (or even the hypothesis of the author) are formulated. Overall, it is very clear to grasp understanding of the manuscript and content in its current state. I strongly advise using hypothesis points to articulate and/or express material in scientific writing. Publication of this piece seems likely in any reputable scientific periodical after a correction in the writing of the manuscript.

In Introduction, I stated research hypotheses, as well as extended the analysis of the results. In Introduction and Conclusions I indicated possible applications of the results of my research.

Table 1-2 is important to explore the specifics. Author needs to wide it. Some conclusions can contribute to the study of the problem.

I extended the characteristics of the banks in the sample, also taking into account the data items added. Additionally I extended the discussion of results included in the Table 2.

Author needs to add more details on the range of simulation considered in this work should be clearly outlined within the abstract. The current statements are vague and too general to get an idea of the work that have been accomplished.

I modified the description of the research methodology, the simulation procedure, and also described the idea of simulation in the Abstract.

The paper possesses a proper form of well-structured and readable technical language of the field and represents the expected knowledge of the journal`s readership. There are minor errors in English, but this does not affect the general nature of the work. The current study brings many new to the existing literature or field. For one, the author(s) seem to have a good grasp of the current literature on their topic area (i.e., recent literature and seminal texts relevant to their study is not cited/referenced).

I significantly extended the scope of the literature, using new items to present the economic background as well as to discuss the results and conclusions of the study.

Round 2

Reviewer 2 Report

This is a nice paper that now could be published.

Author Response

Good day,

I thank you very much for the positive review.

Best regards

Reviewer 3 Report

Dear Authors,

Thanks for your revisions. Even if you added the hypothesis in the introduction, it is not presented appropriately. How can you provide the hypothesis even before reviewing the literature? The Literature review comes after the introduction. I don't think you accept my revision suggestions. My decision has not changed.  

Thanks,

Author Response

I would like to sincerely thank you for your valuable suggestions.

I agree that I misread the hypothesis suggestion. It is logical and fully justified to formulate hypotheses after analyzing the literature.

Correcting this imperfection in the article, I included research hypotheses after conducting a literature review.

Best regards.
